# SentiStream: A Co-Training Framework for Adaptive Online Sentiment Analysis in Evolving Data Streams

**Yuhao Wu[1], Karthick Sharma[2][*], Chun Wei Seah[1], Shuhao Zhang[1][†]**

[1]Singapore University of Technology and Design; [2]University of Sri Jayewardenepura

{yuhao_wu, chunwei_seah, shuhao_zhang}@sutd.edu.sg

en93899@sjp.ac.lk

## Abstract

Online sentiment analysis has emerged as a crucial component in numerous data-driven applications, including social media monitoring, customer feedback analysis, and online reputation management. Despite their importance, current methodologies falter in effectively managing the continuously evolving nature of data streams, largely due to their reliance on substantial, pre-existing labelled datasets. This paper presents SentiStream, a novel co-training framework specifically designed for efficient sentiment analysis within dynamic data streams. Comprising unsupervised, semi-supervised, and stream merge modules, SentiStream guarantees constant adaptability to evolving data landscapes. This research delves into the continuous adaptation of language models for online sentiment analysis, focusing on real-world applications. Experimental evaluations using data streams derived from five benchmark sentiment analysis datasets confirm that our proposed methodology surpasses existing approaches in terms of both accuracy and computational efficiency[1].

## 1 Introduction

Online Sentiment Analysis (OSA) has established its significance in the realm of sentiment analysis, with its primary objective being the identification of polarity in ceaselessly incoming data streams (Capuano et al., 2021). This task requires proficiency in two key aspects: online adaptation and sentiment classification.

The requirement for online adaptation arises from the ever-evolving characteristics of real-time data streams, an effect commonly known as concept drift (Webb et al., 2016).

This necessitates continuous model adaptation to maintain effectiveness. Simultaneously, sentiment classification, a core task within natural language processing (NLP), has found its cruciality in a myriad of sectors such as customer feedback interpretation and public opinion monitoring (Zhang et al., 2018). However, creating an OSA approach that can simultaneously handle online adaptation and sentiment classification remains a challenging feat.

Previous research (Vashishtha and Susan, 2019; Rahnama, 2014; Smailović et al., 2014; Go et al., 2009; Gautam and Yadav, 2014; Haque et al., 2018; Ortigosa et al., 2014) has indicated that supervised learning paradigms can yield high accuracy in sentiment analysis. Despite their strengths, these methods frequently overlook the ceaseless accumulation of real-world streaming data originating from varied sources like literature (Zhu et al., 2015), news articles (Zellers et al., 2019), and scientific papers (Lo et al., 2020). The constant emergence of this dynamic streaming data often leads to the concept drift effect, which can impair the performance of traditional offline methods (Luu et al., 2021). Continuous learning attempts to tackle concept drift and adapt to the ongoing data stream, but obtaining ground truth labels for this streaming data is often arduous and costly, which consequently limits the continuous application of supervised techniques and reduces their long-term efficacy.

In response to these challenges, we present SentiStream, a co-training framework tailored explicitly for efficient online sentiment analysis of swift-flowing opinion data. This framework consists of three modules: unsupervised, semi-supervised, and stream merge. The unsupervised module uses continuously adapted pre-trained language models (PLMs) to distill knowledge from unlabeled streaming data, coupled with lexicon-based strategies to produce preliminary

---

[*]Work done while the second author was interning at SUTD IntelliStream Group.

[†]Corresponding author.

[1]Our code is available at https://github.com/intellistream/SentiStream

polarity labels. `SentiStream` harnesses semantic and temporal information from text-based data streams to incrementally retrain the PLMs, later employing a nimble lexicon-based classification method to generate polarity labels using the updated PLMs. The semi-supervised module constructs a weakly supervised classification model with a small labeled dataset and continuously retrains this model with pseudo-labels generated by the stream merge module. The final stream merge module consolidates outputs from the preceding two modules, utilizing their confidence scores to dynamically update the lexicon for the unsupervised module, providing pseudo-labeled data for semi-supervised learning and dynamically fine-tuning the threshold for the semi-supervised module.

We assessed the performance of SentiStream on five benchmark sentiment analysis datasets and juxtaposing its efficacy against several unsupervised and semi-supervised benchmarks. The experimental results confirm that our methodology considerably outperforms existing methods in tackling dynamic data streams for online sentiment analysis tasks. Additionally, SentiStream uses a lightweight model, ensuring superior throughput and latency performance.

The major contributions of our work can be summarized as:

- The development of `SentiStream`, a novel co-training framework, devised specifically for proficient online sentiment analysis within dynamic data streams;
- An unsupervised module that amalgamates the merits of continuously trained PLMs with lexicon-based classification techniques;
- The implementation of a semi-supervised self-learning strategy, devised to optimize the usage of limited labelled data;
- The unification of outputs through a stream merge technique, promoting collaborative learning to continuously adapt to dynamic stream data from various angles;
- The employment of lightweight models, complemented by a series of optimizations, to fulfill online deployment requirements.

## 2 Related Work

In this section, we provide an overview of relevant literature on online sentiment analysis, continual learning, and semi-supervised learning, thereby laying the foundation for our proposed framework.

### 2.1 Sentiment Analysis

Online sentiment analysis has gained traction with the escalating volume of user-generated content on social media platforms and online forums. MoodLens, a system developed by Zhao et al. (2012), utilizes incremental learning to navigate sentiment shifts and new terminology. In the realm of online text messages, Fernández-Gavilanes et al. (2016) introduced an unsupervised methodology, leveraging sentiment features from lexicons. For a more comprehensive approach, Iosifidis and Ntoutsi (2017) employed semi-supervised learning, drawing upon both labelled and unlabeled data, via Self-Learning and Co-Training. The research in offline sentiment analysis, particularly the remarkable results attained through deep learning architectures like CNNs and RNNs (Kim, 2014; Zhang et al., 2015), and the contributions of pretrained language models such as BERT and GPT (Devlin et al., 2018; Radford et al., 2018), are also worth noting. However, these offline methodologies often falter in adapting to the dynamic nature of online data streams, thereby compromising their performance. This emphasizes the need for specialized online sentiment analysis methods that can adapt fluidly to evolving data streams while maintaining performance levels.

### 2.2 Continual Learning

In contrast to traditional neural networks, which are viewed as static knowledge entities prone to catastrophic forgetting when knowledge expansion efforts veer off the original task, continual learning envisions networks capable of accruing knowledge across different tasks without requiring comprehensive retraining (De Lange et al., 2021). Prior research has primarily sought to address continual learning challenges within the frameworks of incremental class and task scenarios. The strategies used range from replay methods (Rebuffi et al., 2017; Lopez-Paz and Ranzato, 2017; Atkinson et al., 2018) and regularization-based techniques (Kirkpatrick et al., 2017; Ahn et al., 2019) to parameter isolation methods (Xu and Zhu, 2018; Fernando et al., 2017). In a noteworthy contribution, Jin et al. (2021) deployed distillation-based techniques for the continuous incremental pre-training of language models across diverse domain corpora. In the context of sentiment analysis, Ke et al. (2021) explored aspect-based sentiment analysis tasks across different domains

through contrastive continual learning. However, these approaches often fail to address the temporal influence, as observed by Luu et al. (2021), where data drift over time can negatively impact model performance.

## 2.3 Semi-Supervised Learning

Semi-supervised learning, which involves model building using both labelled and unlabeled data, is of particular relevance in real-world scenarios where unlabeled data is plentiful and readily available, while labelled instances are relatively scarce (Ouali et al., 2020). Internet tweets and comments are prime examples that can greatly benefit from semi-supervised learning techniques (Silva et al., 2016). These techniques comprise a variety of methods such as graph-based (Sindhwani and Melville, 2008), wrapper-based (Li et al., 2020), and topic-based (Xiang and Zhou, 2014). More recent studies have delved into dynamic thresholds for semi-supervised techniques (Sohn et al., 2020; Wang et al., 2022), with Han et al. (2020) applying these methods to sentiment analysis. This research suggests an iterative auto-labelling process anchored in a dynamic threshold algorithm, which takes into account both the quality and quantity of auto-labelled data when setting thresholds for their selection.

## 3 Proposed Methodology

Our proposed method, `SentiStream`, consists of two parallel sentiment classification modules and a shared output co-trainer: an unsupervised module, a semi-supervised module, and a stream merge module. The overall structure is shown in Figure 1.

**Problem Formulation:** Our focus is on effectively conducting sentiment analysis of streaming opinion data in real-time. We define the term *input stream* as a sequence of tuples, referred to as $S = T_1, ..., T_N$, which arrive at our system in chronological order. Each tuple, denoted as $T$, consists of a finite number of sentences, $x_i$, forming $T = (x_1 \sim x_m)$. The sentiment polarity is either positive or negative. Our goal with `SentiStream` is to learn and identify the polarity of $T(x_1 \sim x_m) \in S$ as soon as $T_i$ arrives.

## 3.1 Unsupervised Module

Due to potential delays between training cycles, the system might not be up-to-date with events and emerging knowledge (Bubeck et al., 2023). To combat this, we employ two elements: continual pre-trained language model (PLM) training and a dynamic lexicon-based classifier.

**Continuous PLM Training:** In our framework, the PLMs are subject to continuous training, a distinct departure from offline methods (Agarwal and Mittal, 2016; Haque et al., 2018) where PLMs can quickly fall behind as the vocabulary and polarity models evolve. Such offline methods necessitate periodic re-training with labelled datasets, an approach that our framework sidesteps. Our continual training ensures the PLMs remain updated, and capable of labelling sentences that contain even the most novel vocabulary.

The continual PLM training module leverages rich semantic and temporal information from the streaming textual data to incrementally train the models. Specifically, the streaming data is used to perpetually train the model using the pre-trained loss function in an unsupervised manner. The continuous learning aspect of our model means it can learn and refine sentence representations over time, thereby keeping in step with the evolving nature of the data stream. This unique approach allows the model to adapt more effectively to the dynamic language used in current data streams, ensuring it maintains relevance and accuracy.

**Dynamic Lexicon-based Classifier:** After we've obtained the learned vector representations of each word, we suggest leveraging text similarity measures (Wang and Dong, 2020; Vijaymeena M.K, 2016; Navigli and Martelli, 2019) to infer the sentiment polarity of an input sentence. This is based on a list of reference words with pre-established polarities (Lin and He, 2009), as opposed to relying on supervised training progress for polarity identification.

Our lexicon contains various emotion representations that we vectorize using our model. Subsequently, we conduct a cosine similarity computation (CSC) between the vectorized input and the mean of either the positive or negative reference words. The computed aggregated mean is referred to as $Mean(pos)$ or $Mean(neg)$. Ultimately, by comparing $Mean(pos)$ and $Mean(neg)$, we infer the overall polarity of the input tuple.

However, the reference table may become obsolete over time. To address this issue, we dynamically update the lexicon using sentences from the stream merge module, where the model's

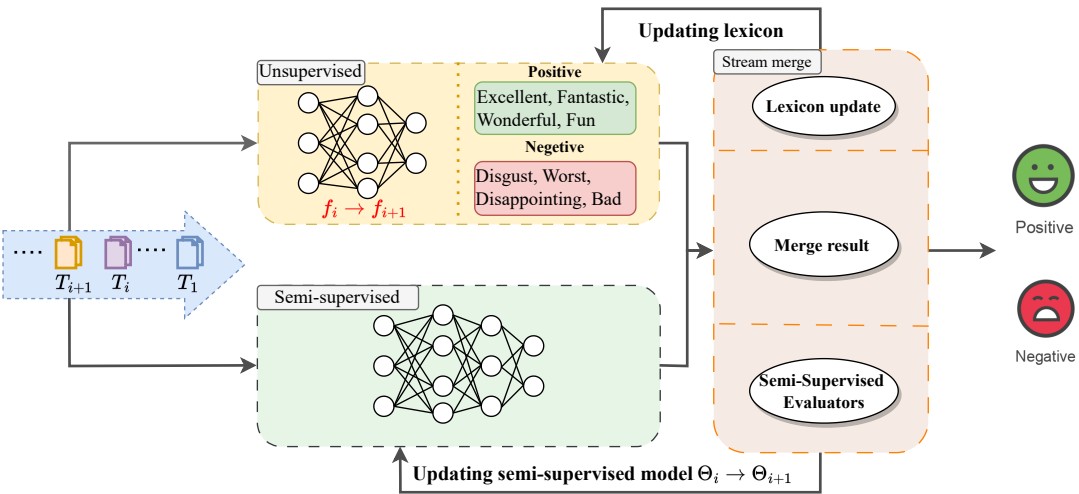

Figure 1: A workflow overview of SentiStream. The unlabelled data enters both the unsupervised and semi-supervised modules, after which the results are merged and output via the stream merge module.

output confidence is high. The lexicon update algorithm, detailed in Algorithm 1, autonomously incorporates new words into the sentiment lexicon, eliminating the need for manual annotation of new data.

We establish a similarity threshold that offers a balance between the need to add new words and the risk of adding false positives to the reference table. Hence, new words are only added to the reference table if the cosine similarity between the lexicon and the word surpasses the similarity threshold.

Through this dynamic update process, our lexicon continuously reflects the prevailing ways emotions are expressed in everyday language. Our extensive experimental results demonstrate that this approach not only significantly reduces computational effort but also consistently outperforms alternative methods in terms of prediction accuracy.

### 3.2 Semi-Supervised Module

The semi-supervised module's primary goal is to efficiently utilize a limited amount of labeled data alongside a substantial amount of unlabeled data obtained from the streaming input, thereby promoting semi-supervised learning. To accomplish this, we use labelled data collected from the stream merge module to identify instances with high confidence, which are then used as pseudo-labels for the continual training of the semi-supervised classifier.

However, the ever-evolving nature of streaming data environments can make the acquisition of

---

**Algorithm 1** Lexicon Update

**Input:** Sentences $S = \{s_1, s_2, \ldots s_n\}$, Initial lexicon $D = \{D_{pos}, D_{neg}\}$, Similarity threshold $\alpha$
**Output:** Updated lexicon $D' = \{D'_{pos}, D'_{neg}\}$
   Compute mean embedding of positive and negative lexicon.
   $\mu_{pos} = \frac{1}{|D_{pos}|} \sum_{d \in D_{pos}}, \mu_{neg} = \frac{1}{|D_{neg}|} \sum_{d \in D_{neg}}$
   **for** each sentence $s$ in $S$ **do**
      **for** each word $w$ in $s$ **do**
         Calculate cosine similarity between $w$ and $\mu_{pos}$
         **if** $\cos(w, \mu_{pos}) > \alpha$ **then**
            Add $w$ to $D'_{pos}$
         **end if**
         Calculate cosine similarity between $w$ and $\mu_{neg}$
         **if** $\cos(w, \mu_{neg}) > \alpha$ **then**
            Add $w$ to $D'_{neg}$
         **end if**
      **end for**
   **end for**

   **return** $D' = \{D'_{pos}, D'_{neg}\}$

---

accurate and consistent pseudo-labels a challenging task. If changes occur within the dataset, it may be difficult for weakly supervised models to make accurate decisions. Likewise, static thresholds may not yield an adequate number of pseudo-labelled data under these changing conditions.

**Dynamic Threshold:** In order to overcome these challenges, we employ a dynamic threshold approach, as proposed in the study by Zhang et al. (2021). As shown in Algorithm 2, this method adjusts the threshold for each class based on the model's current learning status. The learning efficiency of a class is assessed by counting the number of samples whose predictions surpass the hard threshold, defined as the sum of the lower and

upper thresholds. This count is then normalized by the maximum value of either the positive or negative learning effect or the number of low-confidence labels. This approach is especially useful in the early learning stages when the learning effect is generally minimal, and a higher number of low-confidence labels would naturally lead to a more flexible threshold.

Following this, we utilize a non-linear function to adjust the learning rate. Initially, this function incrementally increases the threshold, but it quickens the rise when both learning rates are high. This method allows for a more seamless and logical integration of data, thereby improving the quality of the produced pseudo-labelled data.

---

**Algorithm 2** Dynamic Threshold

---

**Input:** Pseudo labels $P = \{p_1, \ldots p_n\}$, Confidence scores $C = \{c_1, \ldots c_n\}$, Learning effect $\lambda = \{\lambda_{pos}, \lambda_{neg}\}$, Fixed lower threshold $\alpha = \{\alpha_{pos}, \alpha_{neg}\}$, Fixed upper threshold $\beta = \{\beta_{pos}, \beta_{neg}\}$,

**Output:** Filtered pseudo labels $P' = \{p'_1, \ldots p'_m\}$

  $pos \leftarrow \sum_{c > \alpha_{pos} + \beta_{pos}} 1$
  $neg \leftarrow \sum_{c < -(\alpha_{neg} + \beta_{neg})} 1$
  **if** $pos + neg > 0$ **then**
    $\delta \leftarrow \max(|C| - (pos + neg), pos, neg)$
    $\lambda_{pos} \leftarrow (pos/\delta)/(2 - pos/\delta)$
    $\lambda_{neg} \leftarrow (neg/\delta)/(2 - neg/\delta)$
  **end if**
  **for** $c \leftarrow 1$ to $n$ **do**
    **if** $c \leq -(\alpha_{neg} + \beta_{neg} * \lambda_{neg})$ or $c \geq (\alpha_{pos} + \beta_{pos} * \lambda_{pos})$ **then**
      $P' \leftarrow p_c$
    **end if**
  **end for**

  **return** $P' = \{p'_1, \ldots p'_m\}$

---

## 3.3 Stream Merge Module

The principal aim of the stream merge module is to competently amalgamate data and yield reliable pseudo-labelled data. To achieve this, we introduce a stream merge method hinging on *confidence* assessment. As depicted in Algorithm 3, this method merges data streams generated by two separate parts, selecting data points accurately classified (with high confidence) by both models to form pseudo-labels.

The algorithm dynamically adjusts the weights for each model, basing its decision on its previous prediction performance. A model demonstrating a higher ratio of high-confidence predictions will be allocated more weight in the subsequent iteration, thereby potentially improving its prediction accuracy while adapting to the

---

**Algorithm 3** Stream Merge

---

**Input:** Unsupervised model's predicted labels $U_l = U_{l_1}, \ldots U_{l_n}$, Unsupervised model's predicted confidence $U_c = U_{c_1}, \ldots U_{c_n}$, Semi-supervised model's predicted labels $S_l = S_{l_1}, \ldots S_{l_n}$, Semi-supervised model's predicted confidence $S_c = S_{c_1}, \ldots, S_{c_n}$ Fixed confidence threshold $T$, Adaptive weight for unsupervised prediction $W_u$, Adaptive weight for semi-supervised prediction $W_s$

**Output:** Predictions $P' = p'1, \ldots p'm$

  **for** $i \leftarrow 1$ to $n$ **do**
    **if** $Uc_i > T$ and $Sc_i > T$ **then**
      **if** $U_{c_i} > S_{c_i}$ **then**
        $P' \leftarrow U_{l_i}$
      **else**
        $P' \leftarrow S_{l_i}$
      **end if**
    **else**
      **if** $U_{c_i} * W_u > S_{c_i} * W_s$ **then**
        $P' \leftarrow U_{l_i}$
      **else**
        $P' \leftarrow S_{l_i}$
      **end if**
    **end if**
  **end for**
  $W_u \leftarrow \frac{\sum_{i=1}^{n} U_{c_i} > T}{n}$
  $W_s \leftarrow \frac{\sum_{i=1}^{n} S_{c_i} > T}{n}$

  **return** $P' = \{p'_1, \ldots p'_n\}$

---

models' performance over time. For every data point, the algorithm checks whether both models show high confidence in their predictions (surpassing a threshold T). If so, the prediction associated with the highest confidence score is selected. Otherwise, the algorithm multiplies the confidence scores by the adaptive weights for each model, and the prediction with the highest weighted confidence score is chosen.

This approach prioritizes labels with high confidence, which generally lead to correct labels, while also addressing the issue of inaccurate low-confidence labels. By multiplying the model's weight with low-confidence predictions, the algorithm ensures that even when predictions have low confidence, the highest-performing model is given more importance. This, in turn, contributes to making the overall prediction process more consistent and logical.

# 4 Experimental Setup

## 4.1 Datasets

In conducting our evaluation, we employed two distinct types of datasets to thoroughly assess the adaptability of SentiStream to dynamic data streams, each labeled according to specific classification rules. These datasets are delineated as follows:

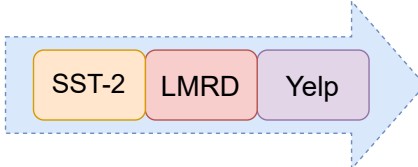

Figure 2: Data stream (Yelp → LMRD → SST-2)

### 4.1.1 Multi-domain Evolving Datasets

Formulated by amalgamating three well-known, large-scale datasets, this construction simulates a dynamic data stream featuring evolving characteristics, thereby reflecting real-world data fluctuations across multiple domains and temporal spans.

**Yelp Review Polarity** (Zhang et al., 2015) was derived from the Yelp Dataset Challenge in 2015. Reviews were labeled as either positive, if they received 3 or 4 stars, or negative, if they received 1 or 2 stars.

**Large Movie Review** (LMRD) (Maas et al., 2011) was collected by the Artificial Intelligence Laboratory at Stanford University. This dataset contains movie reviews along with their associated binary sentiment polarity labels, serving as a benchmark for sentiment classification.

**Stanford Sentiment Treebank-2** (SST-2) (Socher et al., 2013) collected by Stanford University researchers. It consists of movie reviews extracted from Rotten Tomatoes parsed using Stanford parser with sentiment labels.

We evenly sample data from the three datasets, merging them in three different orders (1,Yelp → LMRD → SST-2; 2,LMRD → SST-2 → Yelp; 3, SST-2 → Yelp → LMRD) to simulate real-world data drift, with one of these scenarios illustrated in Figure 2.

### 4.1.2 Longitudinal Singular Domain Datasets

Incorporates two datasets with a longitudinal perspective within a single, consistent domain, providing insight into how SentiStream handles changes and shifts over time within the same data source

**Sentiment140** (Go et al., 2009) was compiled using the Twitter API, encompassing a balanced distribution of 1.6 million tweets, expressed sentiments, and recorded in chronological order from April 6, 2009, to June 25, 2009.

**Amazon Fashion**, a subset of the Amazon Review Data (Ni et al., 2019) and consists of customer reviews for fashion products available on Amazon. These reviews were collected in chronological order in quarterly periods from 2010 to 2018.

## 4.2 Language Model Set

In this paper, we select some lightweight language model to achieve lower latency and higher throughput, in line with the industry's deployment preference for simple and efficient models. Galke and Scherp (2021) showed that combining a bag-of-words model with WideMLP resulted in exceptional performance in text classification tasks. Characterized by a single wide hidden layer, WideMLP outperforms numerous contemporary models in inductive text categorization. Furthermore, training large language models (e.g. BERT, GPT) on streaming data involves additional complexities and constraints, which we will discuss in Section 7. Therefore, here we choose **Word2Vec** (Mikolov et al., 2013) and **Hierarchical Attention Networks** (Yang et al., 2016) as the base models for unsupervised and semi-supervised learning, respectively. We also include **BERT** (Devlin et al., 2018) as a large-scale model for comparative testing.

## 4.3 Baselines

We compare the performance of the proposed model with diverse types of baselines such as random, supervised and self-supervised methods.

- **Random:** At first, we present a random baseline where the predictions are generated using a uniform distribution. This will provide us with a lower bound for our evaluation.

- **Supervised:** We train a supervised model by using 0.5% of the entire dataset as the training set. We chose BERT and HAN as the two base models for our experiments. This will provide us with an upper bound for our evaluation.

- **Self-supervised:** The self-supervised framework used by (Iosifidis and Ntoutsi, 2017) employed co-training to improve the model performance.

- **Weakly-supervised:** We select a weakly-supervised framework (CL-WSTC (Li et al., 2023)) that considers the scenario of continual learning for comparison. They employ BERT as the foundational model.

# 5 Experimental Evaluation

In this section, we study the performance of the different algorithms on three datasets, compare them with different baselines, and discuss the qualitative analysis of our model's performance.

## 5.1 Evaluation Framework

We evaluate our framework within an end-to-end setup for real-time sentiment classification, with throughput, latency, accuracy, and streaming data adaptation serving as primary performance indicators. The task requires processing text data within specific time intervals and assigning the appropriate emotion labels. To verify the effectiveness of our framework, we will run experiments on integrated data streams. We designate 0.5% of the ongoing data stream as training data for our semi-supervised model and other supervised model, while the remaining portion is utilized as test data. This enables us to assess the performance at each stage and the overall performance. An ablation study provides insights into the unique effects of various optimizations, but due to space constraints, this has been relocated to Appendix A.1.

## 5.2 Evaluation Metrics

We evaluate the system with five performance metrics.

**Throughput.** High throughput is a necessity to manage large-volume data streams. For instance, in the event of significant happenings, opinions on social media may suddenly surge. Thus, we measure throughput as the maximum number of input tuples per second that the system can sustain.

**Latency.** We measure the 95% latency as the elapsed time from when the input tuple arrives to when the corresponding classification result is produced. It is an important indicator to denote the system's responsiveness.

**Accuracy.** We define prediction accuracy as the ratio of correct predictions (the sum of *true positives* and *true negatives*) to the total number of tuples processed.

**F1-score & AUC.** To evaluate the accuracy of the prediction in a workload with class imbalance, we also use the *F1-score* and *AUC*, which is the harmonic mean of precision and recall.

| Method | Latency(ms) | Throughput |
|---|---|---|
| Self-supervised | 2.43 | 409 |
| Weakly-supervised | 584.75 | 9 |
| Supervised (BERT) | 53.17 | 162 |
| SentiStream | **0.67** | **1471** |

Table 1: End-to-end evaluation of the throughput and latency

## 5.3 Experimental Results and Analysis

### 5.3.1 End-to-end comparison

**Latency and Throughput:** In Table 1, we can see that SentiStream surpasses self-supervised and weakly-supervised methods, in terms of latency and throughput. The BERT model's latency is nearly 100 times that of SentiStream. Moreover, the difference in throughput is especially pronounced, with SentiStream outperforming the BERT model by nearly an order of magnitude. This discrepancy can be ascribed to the stark contrast in the number of model parameters utilized by SentiStream compared to those used by the BERT model. As for the weakly-supervised method, it was initially developed without considering latency and throughput. Consequently, its approach tends to prioritize future accuracy enhancements, even if it potentially compromises latency and throughput.

### 5.3.2 Multi-domain Evolving Datasets

**Overall Performance:** Table 2 presents a comprehensive summary of our experimental results. Our framework, SentiStream, consistently exhibits excellent performance, occasionally even matching the supervised BERT baseline. In addition, the sentistream approach is always excellent with the supervised HAN and word2vec models. In terms of the F1 score, SentiStream substantially outperforms its counterparts, thereby showcasing the power of its co-training strategy for handling unbalanced data. The performance of the unsupervised component is particularly noteworthy, underscoring the bag-of-words model's ability to produce satisfactory results, even in less complicated text classification tasks. As the data suggests, SentiStream's performance improves with extended offsets.

**Adaptation to Streaming Data:** Figure 3 presents the dynamic performance of accuracy across the entire datasets, assessing the continual learning capability of our method. Initially, during the Yelp dataset stage, supervised learning shows

| Method | Yelp(%) | LMRD(%) | SST-2(%) | Total(%) | F1 | AUC |
|---|---|---|---|---|---|---|
| Random | 49.68 % | 50.12 % | 50.07 % | 50.16 % | 51.07 % | 50.19 % |
| Self-supervised | 64.74 % | 49.81 % | 44.77 % | 57.07 % | 45.74 % | 47.66 % |
| Weakly-supervised | 66.11 % | 51.97 % | 40.91 % | 52.94 % | 42.19 % | 45.11 % |
| Supervised (W2V) | 75.42 % | 60.64 % | 57.65 % | 65.79 % | 65.00 % | 73.44 % |
| Supervised (HAN) | 75.38 % | 65.91 % | 50.24 % | 63.01 % | 68.13 % | 74.85 % |
| Supervised (BERT) | 86.07 % | 78.19 % | 46.48 % | 70.25 % | 72.60 % | 78.99 % |
| Us module [3.1] | 81.60 % | 77.41 % | 76.23 % | 78.79 % | 79.01 % | 82.84 % |
| Ss module [3.2] | 79.80% | 68.63 % | 73.88 % | 74.98 % | 70.49 % | 82.20 % |
| SentiStream | 81.73 % | 77.36 % | 76.22 % | **78.82** % | **79.02** % | **84.97** % |

Table 2: Yelp, LMRD, and SST-2 correspond to the three parts of the merged dataset, Total indicates the average performance of the model in the three orders. F1 refers to the overall F1 score, and the average latency and throughput are also provided. For different methods, the first three are the baselines we have listed in section 4.3, followed by the results for the unsupervised module, semi-supervised module, and our framework in section 3.

remarkable performance, primarily due to the powerful capabilities intrinsic to BERT. However, when concept drift occurs, supervised learning fails to adapt and learn continuously (as it requires annotated data for model fine-tuning), leading to a decline in performance. This downward trend in performance becomes significantly evident during the final task involving the SST-2 dataset.

Surprisingly, methods expected to demonstrate substantial continual learning capabilities, such as self-supervised learning and weakly supervised learning, did not meet the anticipated performance metrics. The underlying model for self-supervised learning is Bayesian, which might be too simple to effectively mine and extract valuable information. Upon closer examination of weakly supervised learning, a major issue was identified: the model's failure to properly integrate seed words during its operation in the online sentiment classification task. This problem is critical given the significant influence these seed words have on the model's performance.

On the contrary, SentiStream displays exceptional performance, consistently adapting to the evolving data distribution inherent to the data stream, thus promoting continuous performance improvement. A prime example is SentiStream's progressively improving performance throughout the SST dataset. It also demonstrated notable resilience in dealing with data drift. Such adaptability is particularly important in real-world scenarios where concept drift can occur in more complex forms.

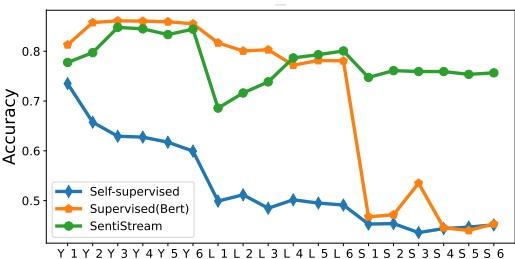

Figure 3: Performance in Multi-domain Stream data, Y is Yelp dataset, L is LMRD dataset, S is SST dataset. Each dataset has six sequential accuracy values, representing the progressive adaptation in the stream data.

### 5.3.3 Longitudinal Singular Domain Datasets

**Overall Performance:** With regard to the aggregate performance, SentiStream exhibits outstanding proficiency on both the Sentiment140 and Amazon Fashion datasets. Notably, SentiStream outperforms its rivals in metrics such as ACC, F1-score, and AUC.

**Adaptation to Streaming Data:** As illustrated in Figures 4 and 5, there are notable performance shifts over time for the self-supervised, supervised, and SentiStream methodologies. In the context of Figure 4, SentiStream markedly distinguishes itself from the other two approaches. On the other hand, in Figure 5, early self-supervised and supervised methods initially lead the pack. However, as the data distribution shifted, for instance during the 2013-Q3 quarter, SentiStream leveraged its superior adaptability and took a comprehensive lead. In summary, SentiStream showcases commendable average performance

| Dataset | Method | ACC(%) | F1 | AUC |
|---------|--------|--------|-----|-----|
| Sentiment140 | Self-supervised | 63.75 % | 63.07 % | 66.26 % |
| | Weakly-supervised | 53.51 % | 52.71 % | 49.23 % |
| | Supervised (W2V) | 66.09 % | 65.78 % | 60.31 % |
| | Supervised (HAN) | 64.87 % | 68.91 % | 61.88 % |
| | Supervised (BERT) | 59.90 % | 66.58 % | 61.96 % |
| | SentiStream | **67.81 %** | **67.56 %** | **72.21 %** |
| Amazon Fashion | Self-supervised | 78.03 % | 87.65 % | 52.07 % |
| | Weakly-supervised | 64.27 % | 51.17 % | 49.95 % |
| | Supervised (W2V) | 78.11 % | 87.71 % | 64.56 % |
| | Supervised (HAN) | 74.81 % | 80.12 % | 69.39 % |
| | Supervised (BERT) | 76.01 % | 82.71 % | 50.00 % |
| | SentiStream | **85.47 %** | **90.03 %** | **93.50 %** |

Table 3: Performance comparison of different methods for Sentiment140 and Amazon (Fashion).

and robust adaptability to streaming data in Longitudinal Singular Domain Datasets.

# 6 Conclusion

This paper introduced SentiStream, an adaptive co-training framework that efficiently tackles concept drift, latency, and throughput issues in dynamic data streams for online sentiment analysis. Through its integrated unsupervised, semi-supervised, and stream merge modules, SentiStream effectively manages continuous data stream evolution, a major challenge for existing methods. Continuous training and dynamic dictionary updates enhance SentiStream's adaptability to ever-changing data streams, proving its potential applicability in real-world scenarios. Experimental results demonstrated SentiStream's promising performance in online sentiment analysis across various data-driven applications. As a highly adaptable and efficient solution, SentiStream addresses the growing demand for real-time sentiment analysis in evolving online environments and data streams. Future work can build on this foundation, extending the application of the SentiStream framework to other dynamic, data-driven domains.

# 7 Limitation

A primary limitation of our study is the lack of integration with popular large-scale language models (e.g., GPT4 (OpenAI, 2023)). However, it is worth noting that employing these models entails increased computational resources and latency. Moreover, current large-scale language models cannot be trained on streaming data, presenting various challenges, such as computational resource utilization, real-time updates, data instability, hyperparameter tuning, storage and management, and system stability and reliability. Bubeck et al. (2023) also emphasizes the importance of continuous learning for LLM and indicates the need for further research and improvement, while our work can be seen as an initial, yet important attempt towards such a goal. Another practical issue is the absence of publicly available corpora categorized by domain or year, along with corresponding sentiment classification test sets. Moving forward, we aim to address this limitation by working with genuine large-scale language models trained on streaming data, showcasing their effectiveness in more complex tasks.

# 8 Acknowledgement

We would like to thank the anonymous reviewers, our meta-reviewer, and senior area chairs for their insightful comments and support with this work. This project is supported by TL@SUTD seed research project grant (RTDSS2214051) and the National Research Foundation, Singapore and Infocomm Media Development Authority under its Future Communications Research & Development Programme FCP-SUTD-RG-2022-006. The computational work for this article was partially performed on resources of the National Supercomputing Centre (NSCC), Singapore (https://www.nscc.sg).

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

# A Appendix

## A.1 Ablation Study

To assess the contribution of each proposed method in improving the performance of our semi-supervised learning model, we conducted an ablation study, as shown in Table 7. The study aimed to evaluate the contributions of dynamic lexicon update and dynamic threshold, as well as their combination, in enhancing sentiment analysis performance. The study consisted of four model variations, namely the baseline model, the model with dynamic lexicon update, the model with dynamic threshold, and the model combining both dynamic lexicon update and dynamic threshold.

By conducting this experiment, we sought to provide empirical evidence regarding the effectiveness of these methods and their impact on sentiment analysis performance. Understanding the contributions of dynamic lexicon update and dynamic threshold, both individually and in combination, can guide the development of more accurate and robust sentiment analysis models, especially in scenarios with limited labeled data availability.

**Baseline Model:** The baseline model served as starting point for comparison. It employed a fixed threshold method with the upper threshold set to 0.8 for filtering pseudo labels and did not incorporate dynamic lexicon update or dynamic threshold.

**Model with Dynamic Lexicon Update:** In this variation, we introduced dynamic lexicon update to the baseline model while maintaining fixed threshold for pseudo label filtering. It is aimed to leverage an evolving lexicon to enhance sentiment classification. Similarity threshold ($\alpha$) was varied to explore its impact on the model's performance. Specifically, we experimented with three values of $\alpha$: 0.7, 0.8 and 0.9.

**Model with Dynamic Threshold:** This variation incorporated dynamic thresholding into the baseline model, but did not involve lexicon updates. It allowed for adaptively adjusting the threshold for pseudo label filtering based on the model's predictions. We examined the effects of different upper threshold values (T) on the model's performance. The same three values of T (0.7, 0.8, and 0.9) used in the previous variation were employed.

**Model with Combined Dynamic Lexicon Update and Dynamic Threshold:** In the final variation, we combined both dynamic lexicon

update and dynamic threshold in the baseline model. This comprehensive approach aimed to leverage the benefits of both techniques simultaneously. For this variation, we fixed the similarity threshold at 0.9 and the upper threshold at 0.8.

The results clearly demonstrate that the combined model outperforms the other variations, indicating the effectiveness of leveraging both dynamic lexicon update and dynamic thresholding for improved sentiment analysis performance.

## A.2 Device Setting

This is information about the device currently used in the experiment.

| CPU | i7-13700ks |
|---|---|
| Memory | 64GB |
| GPU | A6000 |

Table 4: The table shows the device information.

## A.3 Word List

Table 5 shows the initial word list, used in Algorithm 1.

| | |
|---|---|
| Positive | brilliant bliss excellent fantastic super masterpiece admire cool amuse love wonderful best great rejoice beautiful awesome fun |
| Negative | terrible awful unwatchable bad disgust boring stupid bullshit abuse outrage rubbish worst awkward disappointing fraud |

Table 5: Reference table used in our experiments

## A.4 Distribution of Sentiments in Datasets

| Dataset | Negative | Positive |
|---|---|---|
| Yelp | 40227 | 39773 |
| LMRD | 24698 | 24884 |
| SST-2 | 30076 | 37779 |
| Sentiment140 | 800000 | 800000 |
| Amazon Fashion | 170924 | 610225 |

Table 6: Sentiment distribution in datasets

## A.5 Additional Results

### A.5.1 More Training Data Evaluation Results

We used more as (1%) of the total training data, although this is probably the more rare case. In the

| | | Baseline | Lexicon Update | | | Dynamic Threshold | | | Final |
|---|---|---|---|---|---|---|---|---|---|
| | | | $\alpha$=0.7 | $\alpha$=0.8 | $\alpha$=0.9 | T=0.7 | T=0.8 | T=0.9 | |
| Us module | acc | 72.57 % | 61.13 % | 72.65 % | 78.20 % | 71.56 % | 72.40 % | 72.45 % | 78.83 % |
| | f1 | 73.67 % | 68.76 % | 75.87 % | 78.91 % | 72.42 % | 73.45 % | 73.59 % | 79.66 % |
| Ss module | acc | 72.13 % | 68.44 % | 72.13 % | 74.18 % | 71.26 % | 72.92 % | 69.36 % | 74.90 % |
| | f1 | 69.36 % | 63.70 % | 69.96 % | 71.25 % | 63.61 % | 72.57 % | 64.87 % | 71.99 % |
| SentiStream | acc | 72.95 % | 64.82 % | 73.99 % | 77.06 % | 70.98 % | 73.75 % | 70.20 % | 78.94 % |
| | f1 | 71.33 % | 70.46 % | 76.54 % | 77.70 % | 71.24 % | 72.88 % | 66.48 % | 79.60 % |

Table 7: Ablation study of different model variations. The table shows the accuracy and F1 score achieved by each variation under different experimental conditions, such as lexicon update with varying similarity threshold ($\alpha$) and dynamic threshold with varying upper threshold (T).

| Method | Yelp(%) | LMRD(%) | SST-2(%) | Total(%) | F1 | AUC | Latency(ms) | Throughput |
|---|---|---|---|---|---|---|---|---|
| Random | 49.88 % | 50.45 % | 50.26 % | 50.10 % | 50.94 % | 50.06 % | – | – |
| Self-supervised | 68.92 % | 49.83 % | 44.69 % | 55.66 % | 32.19 % | 46.81 % | 2.42 | 480 |
| Supervised | 90.82 % | 81.83 % | 65.98 % | 79.55 % | 79.88 % | 87.83 % | 54.85 | 155 |
| Us module [3.1] | 83.41 % | 76.56 % | 76.16 % | 79.16 % | 80.14 % | 82.96 % | – | – |
| Ss module [3.2] | 82.19% | 68.10 % | 74.00 % | 75.77 % | 75.89 % | 82.22 % | – | – |
| SentiStream | 83.50 % | 76.52 % | 76.00 % | 79.12 % | 80.06 % | 84.98 % | 0.77 | 1292 |

Table 8: The primary distinction between the current table and Table 2 lies in the volume of training data utilized: the present table incorporates 1% more training data, albeit such instances are uncommon. In practical settings, acquiring a substantial amount of labeled data for training, particularly within streaming contexts, is problematic. Real-time manual labeling is virtually infeasible.

results, the supervised effect of BERT is better due to having more training data, but still inevitably followed by a significant drop on the SST-2 dataset as indicated in table 8.

### A.5.2  Multi-domain Evolving Datasets

We conducted comprehensive experiments involving various combinations of datasets to evaluate the performance of our method compared to different baselines. Notably, our method consistently outperformed the alternative baselines in all combinations. Specifically, we examined the Yelp → LMRD → SST-2, LMRD → SST-2 → Yelp and SST-2 → Yelp → LMRD combinations, as illustrated in Table 9, Table 10, Table 11 respectively. The superior effectiveness of our method in addressing the research problem is demonstrated by its consistent performance across various dataset arrangements.

### A.5.3  Longitudinal Singular Domain Datasets

Table 12 and Table 13 present the ACC, F1, AUC, throughput, and latency metrics for longitudinal singular domain datasets. Figure 4 and 5 depict the temporal trends of supervised, semi-supervised, and SentiStream performance on Longitudinal Singular Domain Datasets, specifically Sentiment

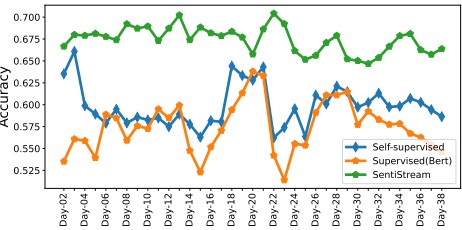

Figure 4: Performance in Longitudinal Stream data (Sentiment 140).

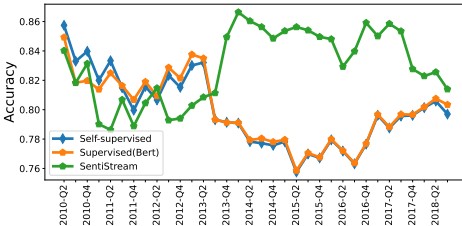

Figure 5: Performance in Longitudinal Stream data (Amazon Fashion).

140 and Amazon Fashion. The SentiStream demonstrates notable average performance and robust adaptability to streaming data within these datasets.

| Method | Yelp(%) | LMRD(%) | SST-2(%) | Total(%) | F1 | AUC |
|---|---|---|---|---|---|---|
| Random | 49.68 % | 50.12 % | 50.07 % | 50.16 % | 51.07 % | 50.19 % |
| Self-supervised | 64.74 % | 49.81 % | 44.77 % | 57.07 % | 45.74 % | 47.66 % |
| Weakly-supervised | 66.11 % | 51.97 % | 40.91 % | 52.94 % | 42.19 % | 45.11 % |
| Supervised (W2V) | 75.42 % | 60.64 % | 57.65 % | 65.79 % | 65.00 % | 73.44 % |
| Supervised (HAN) | 75.38 % | 65.91 % | 50.24 % | 63.01 % | 68.13 % | 74.85 % |
| Supervised (BERT) | 86.07 % | 78.19 % | 46.48 % | 70.25 % | 72.60 % | 78.99 % |
| Us module [3.1] | 81.60 % | 77.41 % | 76.23 % | 78.79 % | 79.01 % | 82.84 % |
| Ss module [3.2] | 79.80% | 68.63 % | 73.88 % | 74.98 % | 70.49 % | 82.20 % |
| SentiStream | 81.73 % | 77.36 % | 76.22 % | **78.82 %** | **79.02 %** | **84.97 %** |

Table 9: Performance comparison of different baselines for Yelp → LMRD → SST-2 combination.

| Method | LMRD(%) | SST-2(%) | Yelp(%) | Total(%) | F1 | AUC |
|---|---|---|---|---|---|---|
| Self-supervised | 73.95 % | 69.08 % | 50.83% | 62.85 % | 53.43 % | 43.78 % |
| Weakly-supervised | 51.48 % | 51.37 % | 43.91 % | 47.59 % | 44.55 % | 43.61 % |
| Supervised (W2V) | 73.79 % | 62.11 % | 72.18 % | 69.02 % | 69.45 % | 74.75 % |
| Supervised (HAN) | 73.14 % | 56.91 % | 74.64% | 70.42 % | 69.16 % | 77.24 % |
| Supervised (BERT) | 80.42 % | 56.17 % | 83.49 % | 73.36 % | 70.16 % | **90.73 %** |
| Us module [3.1] | 76.59 % | 75.69 % | 83.02 % | 78.49 % | 79.83 % | 82.66 % |
| Ss module [3.2] | 74.48% | 74.97 % | 79.86 % | 76.84 % | 78.32 % | 84.03 % |
| SentiStream | 76.63 % | 76.76 % | 83.05 % | **79.34 %** | **80.19 %** | 86.26 % |

Table 10: Performance comparison of different baselines for LMRD → SST-2 → Yelp combination.

| Method | SST-2(%) | Yelp(%) | LMRD(%) | Total(%) | F1 | AUC |
|---|---|---|---|---|---|---|
| Self-supervised | 68.43 % | 49.81 % | 50.12% | 56.23 % | 63.35 % | 48.80 % |
| Weakly-supervised | 55.19 % | 45.57 % | 49.13 % | 49.92 % | 45.64 % | 45.98 % |
| Supervised (W2V) | 55.69 % | 49.72 % | 50.15 % | 51.91 % | 68.35 % | 50.74 % |
| Supervised (HAN) | 53.96 % | 49.81 % | 48.67% | 49.99 % | 68.75 % | 49.79 % |
| Supervised (BERT) | 72.30 % | 60.81 % | 66.12 % | 66.41 % | 71.48 % | 63.58 % |
| Us module [3.1] | 61.13 % | 83.82 % | 81.27 % | 75.40 % | 76.72 % | 75.32 % |
| Ss module [3.2] | 53.21 % | 71.23 % | 67.26 % | 64.04 % | 58.34 % | 75.18 % |
| SentiStream | 61.13 % | 83.88 % | 82.71 % | **75.91 %** | **76.73 %** | **79.38 %** |

Table 11: Performance comparison of different baselines for SST-2 → Yelp → LMRD combination.

| Method | ACC(%) | F1 | AUC | Latency(ms) | Throughput(tuples/s) |
|---|---|---|---|---|---|
| Random | 49.95 % | 40.79 % | 49.95 & | – | – |
| Self-supervised | 63.75 % | 63.07 % | 66.26 % | 3.57 | 412 |
| Weakly-supervised | 53.51 % | 52.71 % | 49.23 % | 419.61 | 19 |
| Supervised (W2V) | 66.09 % | 65.78 % | 60.31 % | 0.01 | 54276 |
| Supervised (HAN) | 64.87 % | 68.91 % | 61.88 % | 0.03 | 17853 |
| Supervised (BERT) | 59.90 % | 66.58 % | 61.96 % | 36.90 | 242 |
| Us module | 67.73 % | 66.46 % | 69.05 % | – | – |
| Ss module | 64.46 % | **67.96 %** | 71.51 % | – | – |
| SentiStream | **67.81 %** | 67.56 % | **72.21 %** | 0.21 | 4518 |

Table 12: Performance comparison of different baselines for Sentiment140.

| Method | ACC(%) | F1 | AUC | Latency(ms) | Throughput(tuples/s) |
|---|---|---|---|---|---|
| Random | 50.06 % | 61.02 % | 50.07 % | – | – |
| Self-supervised | 78.03 % | 87.65 % | 52.07 % | 2.20 | 453 |
| Weakly-supervised | 64.27 % | 51.17 % | 49.95 % | 538.90 | 14 |
| Supervised (W2V) | 78.11 % | 87.71 % | 64.56 % | 0.02 | 43447 |
| Supervised (HAN) | 74.81 % | 80.12 % | 69.39 % | 0.03 | 15132 |
| Supervised (BERT) | 76.01 % | 82.71 % | 50.00 % | 46.07 | 227 |
| Us module [3.1] | 83.02 % | 89.18 % | 87.17 % | – | – |
| Ss module [3.2] | 82.93 % | 86.86 % | 92.29 % | – | – |
| SentiStream | **85.47 %** | **90.03 %** | **93.50 %** | 0.28 | 3511 |

Table 13: Performance comparison of different baselines for Amazon (Fashion).