# OpenReview forum: "SentiStream: A Co-Training Framework for Adaptive Online Sentiment Analysis in Evolving Data Streams"
_EMNLP/2023/Conference — EMNLP 2023 Main_

### Official Review · Reviewer_55iG · 2023-08-02

**Soundness:** 1

**Excitement:**

2: Mediocre: This paper makes marginal contributions (vs non-contemporaneous work), so I would rather not see it in the conference.

**Missing References:**

n/a

**Paper Topic And Main Contributions:**

The paper proposes a streaming strategy for sentiment analysis of online reviews. The proposed model is simple yet seemingly effective. It is a self-correcting pipeline of a supervised and a lexicon-based unsupervised classifier that incrementally updates/corrects itself as the reviews are fed into the model in a temporally ordered stream.

**Questions For The Authors:**

- [resolved if added to the paper] ~~L#380: what are the "specific classifying rules"?~~
- resolved if fixed in the paper~~Figure2: is the caption describe the figure correctly?~~

**Reasons To Accept:**

+ Interesting research direction
+ Well-motivated and well-structured

**Reasons To Reject:**

- The main concern is about the evaluation methodology, which is not sound and includes unclear parts, to my understanding. Specifically:
-- The main dimension of the research, that is time, is not explained. Simply, how are the reviews ordered? What is the time interval? What is the time span of each dataset? ...
-- [resolved if added to the paper] ~~Not sure I understood the "throughput" metric, it's relation with the work, and how it is gauging the efficacy of the proposed model.~~
-- The logic behind mixing the dataset is confusing. How such mixture produce topic drift?
-- No competitive baseline is used. I understand that there might not many works in temporal sentiment analysis, but there are many similar research directions like in temporal recommender systems that could be adopted to benchmark the proposed method against them

-- [resolved] ~~Accuracy is a legit metric in binary classification but in balance datasets. I'm not sure the datasets in the paper's testbed are balanced. There are well-established metrics like auc that could be used to add more reliability for the results.~~

**Reproducibility:**

2: Would be hard pressed to reproduce the results. The contribution depends on data that are simply not available outside the author's institution or consortium; not enough details are provided.

**Reviewer Confidence:**

3: Pretty sure, but there's a chance I missed something. Although I have a good feel for this area in general, I did not carefully check the paper's details, e.g., the math, experimental design, or novelty.

**Typos Grammar Style And Presentation Improvements:**

- not sure the tilde char (~) is a proper choice for showing a stream. I would suggest a list [x1,x2,...xn]
- "using 0.5% of the entire dataset" ==> "50%"?
- "simulating real-world data drift.=> "topic drift" or "semantic drift"

---

> ### Author Rebuttal · Authors · 2023-08-29
>
> **Authors' Response Letter** to the Reviewers of the Manuscript:
>
> SentiStream: A Co-Training Framework for Adaptive Online Sentiment Analysis in Evolving Data Streams
>
> Thanks again for the detailed comments and constructive feedback. Below we discuss how we have addressed each comment as reflected in the revised manuscript. To ease your review, your question will be **bolded**.
>
>
> **Q1: The main dimension of the research, that is time, is not explained. Simply, how are the reviews ordered? What is the time interval? What is the time span of each dataset? The logic behind mixing the dataset is confusing. How such mixture produce topic drift?**
>
> We appreciate the reviewer's inquiry into the time dimension and the rationale behind mixing datasets in our research. We would like to clarify the following:
>
> + *Objective of Dataset Mixing:* The mixing of datasets aims to emulate the complexities of data drift in real-world scenarios. While the datasets are not chronologically ordered, their sequential arrangement captures evolving sentiment expression and language patterns.
> + *Time Periods and Concepts:* Each dataset represents distinct time periods and explores different concepts sequentially. This setup allows us to observe how sentiment expression evolves and adapts to changing contexts.
> + *Examples of Shifts:*
>     + *LMRD & SST:* The transition from LMRD to SST demonstrates a clear evolution in language styles and sentiment lexicons, highlighting the influence of context and domain.
>     + *Yelp & LMRD/SST:* The shift between Yelp and LMRD/SST datasets exemplifies how language and sentiment expressions change across different domains.
>
> + *Additional Experiment:* To further validate our model's performance on real-world datasets, we have included an experiment using Sentiment140 dataset (contains 1.6 million tweets, from 6 Apr 2009 to 25 Jun 2009, with each day representing a distinct time period ) and Amazon-Fashion dataset (contains chronological customer reviews related to fashion products from 2010 to 2018, organized into quarterly periods). Experimental results on a single data stream over a long time span demonstrate the effectiveness of our approach, outperforming other methods.
>
>
> **Sentiment 140 result**
>
> Train dataset: First-day data
>
> Test dataset: Other date data
>
> | Method            | Latency(ms) | Throughput(tuples/s) | ACC   | F1    | AUC   |
> |-------------------|-------------|----------------------|-------|-------|-------|
> | Self-learning     | 3.573       | 212                  | 0.637 | 0.630 | 0.662 |
> | Supervised(Bert)  | 36.898      | 241                  | 0.599 | 0.666 | 0.620 |
> | Us module                | -           | -                    | 0.677 | 0.664 | 0.664 |
> | Ss module             | -           | -                    | 0.671 | **0.679** | 0.715 |
> | **Sentistream**       | **0.208**       | **4674**                 | **0.678** | 0.670 | **0.722** |
>
>
>
> **Amazon (Fashion) result**
>
> Train dataset:  First quarter data
>
> Test dataset:  Other quarter data
>
>
> | Method            | Latency(ms) | Throughput(tuples/s) | ACC   | F1    | AUC   |
> |-------------------|-------------|----------------------|-------|-------|-------|
> | Self-learning     | 2.201       | 452                  | 0.780 | 0.877 | 0.521 |
> | Supervised(Bert)  | 46.056     | 226                  | 0.760 | 0.827 | 0.500 |
> | Us module              | -           | -                    | 0.830 | 0.892 | 0.872 |
> | Ss module             | -           | -                    | 0.829 | 0.869 | 0.923 |
> | **Sentistream**      | **0.279**       | **3511**                 | **0.855** | **0.900** | **0.935** |
>
> **Q2: Not sure I understood the "throughput" metric, it's relation with the work, and how it is gauging the efficacy of the proposed model.**
>
> We appreciate the reviewer's query regarding the "throughput" metric. We would like to clarify the following:
>
> + *Definition:* Throughput quantifies the rate at which our model processes incoming data streams, serving as a measure of analysis speed.
> + *Relevance:* In real-world applications such as online sentiment analysis, where data streams are continuous, rapid analysis is crucial. Thus, throughput is a pivotal metric.
> + *Efficacy Gauge:* By comparing our model's throughput with that of baseline models, we aim to demonstrate its superior capability to handle high data volumes in real-time.
> Throughput, therefore, provides a practical lens through which to assess the real-time applicability and efficacy of our proposed model.
>
> **Q3: No competitive baseline is used. I understand that there might not many works in temporal sentiment analysis, but there are many similar research directions like in temporal recommender systems that could be adopted to benchmark the proposed method against them.**
>
> We appreciate the reviewer's suggestion to consider temporal recommender systems as potential baselines. However, we would like to clarify the following:
>
> + *Domain-Specific Challenges:* While both temporal sentiment analysis and temporal recommender systems involve time-based aspects, the challenges they address are distinct. Our focus is on real-time sentiment classification in dynamic text streams, which differs significantly in granularity, labeling processes, and model complexities from temporal recommender systems.
> + *Choice of Baselines:* Our choice of standard baselines was deliberate, aiming to strike a balance between accuracy and real-time processing speed. Models like LLaMA and ChatGpt, although potentially more accurate, come with computational complexities that may not be suitable for real-time applications.
> + *Competitive Basis:* The selected baselines, such as self-supervised and weakly-supervised methods, represent state-of-the-art solutions that offer a strong competitive basis for our comprehensive comparisons. They effectively address the challenges of concept drift and large-scale sentiment analysis with limited labeled data.
>
> Given these considerations, we believe our choice of baselines is aligned with the specific focus and challenges of our research. We will continue to explore ways to enrich our evaluation methodology while maintaining its applicability to our domain.
>
> **Q4: Accuracy is a legit metric in binary classification but in balance datasets. I'm not sure the datasets in the paper's testbed are balanced. There are well-established metrics like auc that could be used to add more reliability for the results.**
>
> We appreciate the reviewer's suggestion to consider alternative metrics for more reliable results, especially in the context of potentially imbalanced datasets.
> + *Additional Metrics:* In response to your suggestion, we have included supplementary experiments focusing on the AUC metric, as shown in the table below. The experimental demonstration illustrates that Sentistream outperforms other methods in  AUC metrics.
>
> | Method            | AUC    |
> |-------------------|--------|
> | random            | 50.19% |
> | Self-supervised   | 47.66% |
> | Supervised (W2V)  | 73.44% |
> | Supervised (BERT) | 78.99% |
> | Us Nodule      | 82.84% |
> | Ss Module         | 82.20% |
> | Sentistream       | **84.97%** |
>
> + *F1 Score:* We reported the F1 scores in Table 2 of the original paper. The F1 scores are well suited for evaluating the performance of unbalanced datasets.
> + *Dataset Distributions:* Our method demonstrates superior performance in both balanced and imbalanced datasets compared to other approaches. Detailed dataset distributions have been provided to offer context for the evaluation metrics used.
>
> | Dataset           | Negative | Positive |
> |-------------------|----------|----------|
> | Yelp              | 40227    | 39773    |
> | LMRD              | 24698    | 24884    |
> | SST-2             | 30076    | 37779    |
> | Sentiment140      | 800000   | 800000   |
> | Amazon (Fashion)  | 170924   | 610225   |
>
>
> **Q5: L#380: what are the "specific classifying rules"?**
>
> The term "specific classifying rules" pertains to the situation in which the Yelp Review dataset, originally structured with four levels of sentiment polarity (ranging from 1 to 4), was subsequently converted into a two-class classification dataset (1,2 → positive; 3,4 → negative) in our paper. In contrast, LMRD and SST-2 datasets is two-classifications dataset, distinguishing between positive and negative sentiments.
>
>
> **Q6: Figure2: is the caption describe the figure correctly?**
>
>
> The original caption was not accurate, the correct caption is: Data stream (Yelp → LMRD → SST-2).

---

### Official Review · Reviewer_o7C8 · 2023-08-03

**Typos Grammar Style And Presentation Improvements:** 1. "Continual Training" -> "Continuou…
**Soundness:** 3

**Excitement:**

4: Strong: This paper deepens the understanding of some phenomenon or lowers the barriers to an existing research direction.

**Paper Topic And Main Contributions:**

This submission introduces SentiStream, a sentiment analysis module for streaming data. The authors present the components of their architecture in detail and proceed in evaluating it on three datasets against four other baselines, reporting on their findings.

**Reasons To Accept:**

1. The authors research an interesting topic, that of sentiment analysis of data streams.
2. The proposed architecture tries to address key issues of stream data, such concept drift and system latency.

**Reasons To Reject:**

1. In the experimental results, they only compare with some standard baseline and not with the state-of-the-art in sentiment analysis on data streams
2. In the way sentiment polarities are approximated (employing sentiment lexicons and computing mean of positive vs mean of negative words), they miss the context of the review (eg a review text of the form "Even though the restaurant was expensive and the orders were late, I enjoyed the experience" would be classified as negative, even though it is positive).

**Reproducibility:**

4: Could mostly reproduce the results, but there may be some variation because of sample variance or minor variations in their interpretation of the protocol or method.

**Reviewer Confidence:**

4: Quite sure. I tried to check the important points carefully. It's unlikely, though conceivable, that I missed something that should affect my ratings.

---

> ### Author Rebuttal · Authors · 2023-08-29
>
> **Authors' Response Letter** to the Reviewers of the Manuscript:
>
> SentiStream: A Co-Training Framework for Adaptive Online Sentiment Analysis in Evolving Data Streams
>
> Thanks again for the detailed comments and constructive feedback. Below we discuss how we have addressed each comment as reflected in the revised manuscript. To ease your review, your question will be **bolded**.
>
>
> **Q1: In the experimental results, they only compare with some standard baseline and not with the state-of-the-art in sentiment analysis on data streams.**
>
> We appreciate the reviewer's observation regarding the comparison with state-of-the-art methods. We would like to clarify the following:
>
> + *Lack of Universal SOTA:* It is important to note that there is currently no universally accepted state-of-the-art method for sentiment analysis on data streams, making direct comparisons challenging.
> + *Selected Advanced Methods:* Despite this, we have included comparisons with advanced methods in the field. Specifically, we have chosen CL-WSAT[1] for continual learning in text classification tasks, and a self-supervised method[2] applied to large-scale Twitter data streams. In our paper, these are respectively referred to as "weakly-supervised" and "self-supervised."
>
> By including these advanced methods, we aim to provide a more comprehensive evaluation of our approach in the context of existing research.
>
>
> **Q2: In the way sentiment polarities are approximated (employing sentiment lexicons and computing mean of positive vs mean of negative words), they miss the context of the review (e.g. a review text of the form "Even though the restaurant was expensive and the orders were late, I enjoyed the experience" would be classified as negative, even though it is positive).**
>
> We appreciate the reviewer's concern regarding the approximation of sentiment polarities and the potential for missing contextual nuances. We would like to clarify the following points:
>
> + *Scope of Analysis:* Our paper focuses on general sentiment analysis. Targeted sentiment analysis or stance detection, which would require different methodologies, is beyond the scope of this work.
> + *Contextual Understanding:* While we employ Word2Vec, a non-contextual word embedding method, our approach also integrates Hierarchical Attention Networks (HAN) in the semi-supervised module. HAN is designed to capture the context and nuances within the review text effectively.
> + *Trade-offs:* We acknowledge that using more advanced contextual embeddings like BERT could provide richer contextual information. However, in scenarios requiring low latency, such embeddings may not be practical. Our approach aims to balance accuracy and real-time usability. In particular, as shown in Table 1, the BERT model’s latency is nearly 100 times that of SentiStream.
>
> | Method             | Latency(ms) | Throughput(tuples/s) |
> |--------------------|-------------|------------|
> | Supervised (BERT)  | 53.17       | 162        |
> | SentiStream        | **0.67**        | **1471**       |
>
>
> + *Real-time Feasibility:* By combining Word2Vec and HAN, we strive to offer meaningful sentiment predictions with manageable latency, making our approach suitable for online stream environments.
>
>
>
> [1]CL-WSTC: Continual Learning for Weakly Supervised Text Classification on the Internet
>
> [2]Large-scale sentiment learning with limited labels

---

### Official Review · Reviewer_11Lv · 2023-08-05

**Soundness:** 3

**Excitement:**

4: Strong: This paper deepens the understanding of some phenomenon or lowers the barriers to an existing research direction.

**Paper Topic And Main Contributions:**

The paper proposes a co-training method for sentiment analysis on stream data. The proposed framework consists of three modules. The unsupervised module aims at updating the pre-trained language model, which will shift over time. The goal of the semi-supervised module is to update the model based on the pseudo-labels obtained. The stream merge module is designed to combine the prediction results from the unsupervised module and semi-supervised module. Experiments on publicly available datasets were conducted. Comparison with existing baselines shows the proposed method achieves promising results.

**Reasons To Accept:**

1.	The paper is addressing an interesting research problem about the data draft in sentiment analysis. It is of the interest of the general audience of the conference.
2.	The rationale and the design of the proposed framework is elaborated in detail.
3.	Empirical results show that the proposed method outperforms other baselines.


**Reasons To Reject:**

1.	Empirical results show that the unsupervised module is the major source of improvement in the performance. The integration of the semi-supervised module can only slightly improve, or even degrade the overall performance. It cannot justify the need of the semi-supervised module in the framework. In particularly, without the semi-supervised module, the framework does not need any training examples.
2.	Following this, the proposed framework employs lightweight language model in the unsupervised module. If a more sophisticated language model is used, it is unclear if the semi-supervised module is still needed.
3.	Authors may consider using a running example to explain the concept draft of sentiment stream. The experiments use three different dataset to simulate the data stream and sentiment draft. However, an example of concept draft in a single source can strengthen the need of the application.


**Reproducibility:**

4: Could mostly reproduce the results, but there may be some variation because of sample variance or minor variations in their interpretation of the protocol or method.

**Reviewer Confidence:**

4: Quite sure. I tried to check the important points carefully. It's unlikely, though conceivable, that I missed something that should affect my ratings.

---

> ### Author Rebuttal · Authors · 2023-08-29
>
> **Authors' Response Letter** to the Reviewers of the Manuscript:
>
> SentiStream: A Co-Training Framework for Adaptive Online Sentiment Analysis in Evolving Data Streams
>
> Thanks again for the detailed comments and constructive feedback. Below we discuss how we have addressed each comment as reflected in the revised manuscript. To ease your review, your question will be **bolded**.
>
> **Q1: Empirical results show that the unsupervised module is the major source of improvement in the performance. The integration of the semi-supervised module can only slightly improve, or even degrade the overall performance. It cannot justify the need for the semi-supervised module in the framework. In particular, without the semi-supervised module, the framework does not need any training examples. Following this, the proposed framework employs a lightweight language model in the unsupervised module. If a more sophisticated language model is used, it is unclear if the semi-supervised module is still needed.**
>
> We appreciate the reviewer's insightful comment on the role of the semi-supervised module in our framework. We acknowledge that the unsupervised module is a significant contributor to performance. We would like to clarify the following points:
>
> - *Online Setting Adaptability:* Large language models, even when fine-tuned, may not adapt swiftly enough to rapidly changing data streams (Ghunaim et al. 2023). Our semi-supervised module is designed to mitigate this limitation by leveraging a small amount of labeled data for quicker adaptability.
> - *Initial Phase Performance:* During the initial phase, where the unsupervised module has not yet adapted to the data stream, the semi-supervised module demonstrates superior performance. This is crucial for applications where immediate accuracy is imperative. for example,  such as the continuous emergence of new domains (concept drift occurs more frequently and more severely), Ss module can exert a greater impact due to its strong initial performance. As shown in Figure 3 of the original paper, Sentistream is capable of rapidly recovering a stable performance level.
> - *Dataset Proportion:* It's worth noting that the labeled data used for semi-supervised learning constitutes only 0.5% of the total dataset, emphasizing its efficiency.
>
>
>
>
> **Q2: Authors may consider using a running example to explain the concept draft of sentiment stream. The experiments use three different dataset to simulate the data stream and sentiment draft. However, an example of concept draft in a single source can strengthen the need of the application.**
>
> We are grateful for the reviewer's suggestion to include a running example to elucidate the concept of sentiment stream drift. We concur that a single-source example could offer a more focused illustration. Nevertheless, we hope to highlight that:
>
> - *Dataset Limitations:* The use of multiple datasets (SST, LMRD, Yelp) was necessitated by the absence of single, large-timespan datasets that adequately represent concept drift. We have acknowledged this limitation in our manuscript.
> - *Example of Mixed Streams:* We provide an example comparing the transition from the LMRD to the SST dataset to highlight the complexities of mixed sentiment streams. This serves to emphasize the challenges in handling concept drift across diverse domains.
> - *Future Work:* To directly address your suggestion, we plan to include an experiment using the Sentiment140 dataset (contains 1.6 million tweets,  from 6 Apr 2009 to 25 Jun 2009, with each day representing a distinct time period) and Amazon-Fashion dataset (contains chronological customer reviews related to fashion products from 2010 to 2018, organized into quarterly periods). Preliminary results indicate that our approach outperforms existing methods in terms of accuracy, F1 score, and AUC, while maintaining low latency and high throughput in the below table. Experimental results on a single data stream over a long time span demonstrate the effectiveness of our approach, outperforming other methods.
>
> **Sentiment 140 result**
>
> Train dataset: First-day data
>
> Test dataset: Other date data
>
> | Method            | Latency(ms) | Throughput(tuples/s) | ACC   | F1    | AUC   |
> |-------------------|-------------|----------------------|-------|-------|-------|
> | Self-learning     | 3.573       | 212                  | 0.637 | 0.630 | 0.662 |
> | Supervised(Bert)  | 36.898      | 241                  | 0.599 | 0.666 | 0.620 |
> | Us module              | -           | -                    | 0.677 | 0.664 | 0.664 |
> | Ss module             | -           | -                    | 0.671 | **0.679** | 0.715 |
> | **Sentistream**       | **0.208**       | **4674**                 | **0.678** | 0.670 | **0.722** |
>
>
>
>
>
> **Amazon (Fashion) result**
>
> Train dataset: First quarter  data
>
> Test dataset: Other quarter data
>
> | Method            | Latency(ms) | Throughput(tuples/s) | ACC   | F1    | AUC   |
> |-------------------|-------------|----------------------|-------|-------|-------|
> | Self-learning     | 2.201       | 452                  | 0.780 | 0.877 | 0.521 |
> | Supervised(Bert)  | 46.056     | 226                  | 0.760 | 0.827 | 0.500 |
> | Us module                | -           | -                    | 0.830 | 0.892 | 0.872 |
> | Ss module              | -           | -                    | 0.829 | 0.869 | 0.923 |
> | **Sentistream**       | **0.279**       | **3511**                 | **0.855** | **0.900** | **0.935** |

---

### Meta-Review · Area_Chair_yP8w · 2023-09-15

**Recommendation:** 3

**Metareview:**

The problem statement, solution, methods, and experiments are clearly explained. The manuscript is well-written and it is very easy to follow. The research topic is interesting and although the individual methods are not novel, the way the methods are combined together, the overall solution is interesting.

However, one of the major contributions of the work is to build a model that can be adapted to the dynamic of the data, however, the dataset they used does not completely capture the capabilities of the framework they are proposing. The datasets are collected in different years and they have different genres certainly, they are different in nature, but one cannot compare the nature of a static dataset in a specific era of time with the nature of the data stream.

In addition, another concern is the way sentiment words are added to the seed words with positive and negative polarity. Sentiment is such a subjective phenomenon and it is very hard to add words based on a threshold since words in different contexts could mean differently. This matter is also mentioned by one of the reviewers.

Lastly, all the baseline and other methods that are compared with this framework are static and that is not a fair comparison because these models are static.

---

### Decision · Program_Chairs · 2023-10-07

**Decision:**

Accept-Main

**Comment:**

The problem statement, solution, methods, and experiments are clearly explained. The manuscript is well-written and it is very easy to follow. The research topic is interesting and although the individual methods are not novel, the way the methods are combined together, the overall solution is interesting.

However, one of the major contributions of the work is to build a model that can be adapted to the dynamic of the data, however, the dataset they used does not completely capture the capabilities of the framework they are proposing. The datasets are collected in different years and they have different genres certainly, they are different in nature, but one cannot compare the nature of a static dataset in a specific era of time with the nature of the data stream.

In addition, another concern is the way sentiment words are added to the seed words with positive and negative polarity. Sentiment is such a subjective phenomenon and it is very hard to add words based on a threshold since words in different contexts could mean differently. This matter is also mentioned by one of the reviewers.

Lastly, all the baseline and other methods that are compared with this framework are static and that is not a fair comparison because these models are static.